# The Role of Positron Emission Tomography and Computed Tomographic (PET/CT) Imaging for Radiation Therapy Planning: A Literature Review

**DOI:** 10.3390/diagnostics13010053

**Published:** 2022-12-24

**Authors:** Abba Mallum, Thokozani Mkhize, John M. Akudugu, Wilfred Ngwa, Mariza Vorster

**Affiliations:** 1Department of Radiotherapy and Oncology, College of Health Sciences, University of KwaZulu-Natal, Durban 4000, South Africa; 2Department of Radiotherapy and Oncology, Inkosi Albert Luthuli Central Hospital, Durban 4091, South Africa; 3University of Maiduguri Teaching Hospital, Maiduguri 600104, Nigeria; 4Department of Nuclear Medicine, College of Health Sciences, University of KwaZulu-Natal, Durban 4000, South Africa; 5Department of Nuclear Medicine, Inkosi Albert Central Hospital, Durban 4091, South Africa; 6Division of Radiobiology, Department of Medical Imaging and Clinical Oncology, Faculty of Medicine and Health Sciences, Stellenbosch University, Tygerberg 7505, South Africa; 7School of Medicine, Johns Hopkins University, Baltimore, MD 21218, USA; 8Brigham and Women’s Hospital, Dana-Farmer Cancer Institute, Harvard Medical School, Boston, MA 02115, USA

**Keywords:** radiotherapy, PET/CT treatment planning system

## Abstract

PET/CT is revolutionising radiotherapy treatment planning in many cancer sites. While its utility has been confirmed in some cancer sites, and is used in routine clinical practice, it is still at an experimental stage in many other cancer sites. This review discusses the utility of PET/CT in cancer sites where the role of PET/CT has been established in cases such as head and neck, cervix, brain, and lung cancers, as well as cancer sites where the role of PET/CT is still under investigation such as uterine, ovarian, and prostate cancers. Finally, the review touches on PET/CT utilisation in Africa.

## 1. Introduction

Radiotherapy (RT) is a crucial modality of cancer treatment. About two-thirds of all cancer patients in Western countries receive radiotherapy as a curative modality, either alone or in combination with other cancer treatment modalities [1]. Radiotherapy has advanced tremendously over the last century. Along with the evolution and advancement of radiotherapy, there has been an advancement in medical imaging techniques, which play a crucial role in the planning and delivery of radiation therapy.

Radiation therapy planning can be defined as the process of image acquisition, volume delineation, dose-fractionation prescription, assigning of treatment fields and beam modifiers, evaluation of dose distribution, and quality assurance before final approval for treatment delivery [2]. Radiotherapy treatment planning is complex and relies heavily on imaging and computing technologies to ensure the delivery of therapeutic doses of radiation to the tumour while minimising the amount of radiation to the adjacent healthy tissue [3].

Since the 1990s, there have been rapid technological advancements in anatomical imaging modalities, such as computed tomography (CT) and magnetic resonance imaging (MRI), with innovations such as image-guided radiotherapy (IGRT), intensity-modulated radiotherapy (IMRT), stereotactic body radiotherapy (SBRT), and proton beam therapy (PBT) [4].

Positron emission tomography (PET) is an advanced functional imaging modality. Although it is mainly used for the diagnosis, staging, prognostication, and surveillance of numerous types of oncology care, PET imaging can be used in radiotherapy treatment planning. CT or MRI-based imaging are currently the main radiological techniques used, but PET imaging is increasingly being integrated within RT planning [5,6]. A major advantage of PET functional imaging is its ability to distinguish between neoplastic and normal tissues with more accuracy than CT or MRI, which are reliant on morphological features to make this differentiation [6,7].

While more scientific research and clinical trials are ongoing on this subject, we provide an overview of the current state of use of PET/CT in treatment planning in different cancer types.

## 2. Principles of ^18^F-FDG-PET

The basic principle of PET is that cyclotron-generated positron-emitting radiopharmaceuticals are administered to a patient intravenously. The isotopes decay to emit a neutron, a positron (positively charged electrons sometimes called a β^+^ particle) and a neutrino. The positrons annihilate with electrons to produce two 511-keV photons directed approximately 180° apart. The external detectors on the PET scanner detect these photons [8,9].

Radiopharmaceuticals (tracers) are biologically important materials such as glucose or oxygen, which have been labelled with radionuclides such as ^11^Carbon, ^13^Nitrogen, ^15^Oxygen and ^18^Fluoride [10]. There are many types of tracers used in PET scans (and more under investigation) depending on the organ of interest; however, ^18^F-labelled fluoro-2-deoxyglucose (^18^F-FDG) is the most common tracer widely used [11]. The tracer ^18^F-FDG tracer is a glucose analogue which enters the cells via glucose transporters. It is taken up more avidly by tumours than healthy tissues due to the increased rate of glucose metabolism in tumours, and it becomes trapped inside tumour cells due to the fluorine substitution in the glucose molecule [8,9]. Other tracers that depict other tumour characteristics apart from glucose metabolism include ^18^F-Fluoro-L-dihydroxyphenylamine (^18^F-fluoro-L-DOPA), Somatostatin-based radiotracers, ^11^C-Choline, ^18^F-16 β-Fluoro-5α-dihydrotestosterone (FDHT), ^18^F-3-Fluoro-3-deoxy-thymidine (^18^F-FLT), ^11^C-Acetate, ^18^F-Fluoride, and ^11^C-Methionine [10].

## 3. Role of PET in Radiotherapy Treatment Planning

PET can be used in various ways for radiation therapy planning, including as a visual aid for target delineation, using a fusion of PET and CT images acquired from separate scanners, or by obtaining the images on an integrated PET/CT unit while the patient is in the treatment position [2]. Integrated PET/CT units [See Figure 1 below] simplify image registration in treatment planning because both anatomical and metabolic images are acquired with the patient in the same position and during a single procedure [2].

A distinct advantage of PET/CT is its ability to improve tumour delineation, which reduces both intra-observer and inter-observer variation in tumour delineation, thereby reducing the chance of geographically missing part of the tumour during the treatment [2,12]. Furthermore, by identifying functional sub-volumes, the metabolic data from a PET scan can be used better to describe the anatomical gross tumour volume (GTV), thereby defining a biological target volume (BTV) [11]. ‘Biological target volume’ is a concept introduced by Ling et al. in 2020, and it represents the sub-volume of the tumour with specific characteristics of functional or molecular imaging techniques [13]. The BTV paved the way for the concept of “dose painting”, in which the target volume receives an inhomogeneous dose distribution based on its functional activity. Two distinct dose-painting strategies, dose escalation and dose redistribution, have been proposed. Dose redistribution consists of increasing the dose to the radioresistant sub-volumes while decreasing the dose to the rest of the target. This strategy is appealing because it could theoretically increase the dose to targets deemed to require higher doses while decreasing the dose to vital organs and reducing the chances of late radiation-induced toxicity [2,11,12].

PET/CT has been used in the target volume definition of many cancer sites, including the brain, head and neck, lungs, prostate, gynaecological, anal, and other cancers.

### 3.1. Central Nervous System (Brain) Cancers

Primary CNS cancers are rare. In 2020, 308,000 cases of cancer of the central nervous system and 251,000 deaths from this cancer were reported globally [14]. Gliomas are the second most common primary tumour of the brain and the second leading cause of cancer mortality in adults under 35 years of age [15].

Because of the brain’s vulnerability, accurate identification of the treatment target is crucial. Target-volume delineation for radiotherapy planning is currently based on CT and MRI, which is considered the gold standard [11,16]. However, conventional MRI sequences are limited in their ability to differentiate between oedema, non-enhancing and infiltrating tumours, enhancing tumours in gliomas, and inadequately assessed tumour margins in non-enhancing lesion [16,17].

In the application of PET in neuro-oncology, non-glucose tracers have been extensively used and have shown clear superiority over MRI [5,11,16]. Although ^18^F-FDG is the most common radiopharmaceutical in clinical nuclear oncology, this tracer has some intrinsic limitations due to the physiological high glucose metabolism within the healthy brain parenchyma, resulting in poor tumour-to-background contrast [11,15,17]. In gliomas, radiolabelled amino acid tracers have been used for brain tumour PET imaging because of their increased uptake in the neoplastic tissue but low uptake in healthy brain parenchyma. Their increased uptake is related to increased transport via amino acid the transporters of the L-type (LAT); the LAT subtypes 1 and 2 are overexpressed in neoplastic tissue [11]. These amino acid tracers include ^11^C-methyl-L-methionine (^11^CMET), O-(2-^18^F fluoroethyl)-L-tyrosine (FET), and 3,4-dihydroxy-6-^18^F-fluoro-L-phenylalanine (FDOPA) [17,18,19,20]. The overexpression of somatostatin receptors (SSTR) in meningiomas allows for the use of somatostatin-based radio tracers in PET for meningiomas. DOTA-D-Phe1-Tyr3-octreotate (DOTATATE) and DOTA-Tyr3-octreotide (DOTATOC) are the most common tracers in the PET for meningiomas and patients with pituitary and neuroendocrine tumours [15,17,18,19].

Studies have suggested that MRI alone significantly underestimates the metabolically active tumour volume, making it essential to complement MRI with PET scans in treatment planning for brain tumours [19,21,22]. Indeed, Galldicks et al., in a report of the PET/RANO Group in 2021, recommended that the most frequently used radiolabelled amino acid tracers ^11^C-methyl-L-methionine(^11^CMET), O-(2-^18^F-fluoroethyl)-L-tyrosine (FET), and 3,4-dihydroxy-6-^18^F-fluoro-L-phenylalanine (FDOPA) may improve the delineation of radiotherapy target volumes beyond conventional MRI and identify additional tumour parts that should be targeted by irradiation (level 2 evidence) [17]. FDOPA PET can also be used to differentiate a glioma from other benign tumours in the brain. While a PET scan is an essential component of treatment planning, MRI remains the gold standard.

### 3.2. Head and Neck Tumours

PET/CT is routinely used in different aspects of the management of patients with head and neck cancers, including treatment planning as summaries on Table 1 below.

The main tracer in head and neck PET is FDG. Farré et al., in 2013, in a study of 22 patients with stage II-IV squamous cell carcinoma of the head and neck who underwent ^18^FCG-PET/CT in the treatment positioning for staging and planning, showed that ^18^FCG-PET/CT improved patient selection and provided helpful information in treatment planning, and does not increase workload. It is cost-effective for the patients selected for radiotherapy [23]. Several studies reported that ^18^F-FDG PET/CT might alter the initial radiotherapy planning in up to one-third of patients with higher accuracy. Daisne et al., in 2004, in a study of 29 patients, compared gross tumour volumes (GTV) delineated from CT, MRI and PET-CT images. They showed that contouring with PET/CT yielded a sharply defined GTV, which was significantly smaller than those from contrast-enhanced CT or MRI [24]. Among nine patients with surgical specimens available, PET was found to be the most accurate modality [24]. This finding was corroborated by other studies [25,26,27].

The prognostic relevance of ^18^FCG-PET/CT was evaluated in a systematic review and meta-analysis by Bonomo et al. in 2018. The study conducted a systematic review of the literature and a meta-analysis of 25 studies, with a total of 2223 patients. The authors concluded that metabolic tumour volume (MTV) was defined from pre-treatment.

^18^FDG PET scans have the strongest impact on patient outcomes after standard concurrent chemoradiotherapy, therefore, ^18^FDG PET has prognostic relevance in the context of locally advanced head and neck squamous cell carcinoma [28].

Most relapses in patients with locally advanced head and neck cancer are locoregional. It is widely acknowledged that increasing the radiation dose can enhance local control because there is a dose-response relationship. To this effect, Michaelidou et al. carried out a phase 1 dose-escalating study (FiGaRO trial) to establish the safety and feasibility of using an ^18^FDG-PET-baseddose-painting technique to deliver a radiotherapy (RT) boost to the ^18^FDG-avid primary tumour in patients with locally advanced high and intermediate risk oropharyngeal cancer [29]. The study, which included 24 patients, concluded that it was feasible to deliver PET/CT guided escalated IMRT across centers in the UK, and that late toxicity rates were comparable to standard dose chemo-IMRT. The results of the study also suggested an improved 3-year survival rate for high-risk patients [29].

Because areas of hypoxia correlate with locoregional treatment failure rates, tracers that provide quantitative measurements of tissue hypoxia, ^18^F-fluoromisonidazole (FMISO) and ^18^F-fluoroazomycin arabinoside (FAZA), are being investigated.

Under clinical investigation are an imaging biomarker for tumour cell proliferation (^18^FFLT) and a new imaging biomarker (FAPI) for a new biological target, the fibroblast activation protein (FAP) [5]. While PET/CT is currently used in managing head and neck cancers, large clinical trials are warranted to establish the exact position of PET/CT in the radiotherapy of head and neck cancer patients.

### 3.3. Lung Cancer

The tracer ^18^F-FDG-PET/CT is the recommended imaging modality for lung cancer staging. According to the NCCN, ESMO, ESTRO, ACROP and EORTC recommendations, FDG-PET/CT plays a crucial role in RT planning in lung cancer for both primary tumours and lymph node metastases [30].

In lung cancer treatment planning, PET/CT is reported to have high sensitivity and specificity for detecting distant metastases, allowing for accurate staging and treatment allocation. PET/CT-based target volumes are more uniform and correlate more accurately with surgical pathology specimens for operative cases. It also ensures adequate dose coverage for areas potentially missed by CT alone. It is valuable in identifying tumour boundaries in case of extra-thoracic or mediastinal tumour extension, when the tumour and normal tissue have similar visual appearance on CT, as in cases of atelectasis caused by compression of the airways by tumour, or in cases where there is significant fibrosis of the lung [5,7,11,30].

Hallqvist et al. in 2017, published a systematic review and meta-analysis involving 36 studies, which evaluated whether the use of PET/CT for dose planning purposes improves radiotherapy for patients suitable for curative treatment with high dose radiochemotherapy, and to quantify the effect on patient selection and target definition [31]. Their report indicated that approximately two out of five patients will have a significant change in target definition, and one out of five patients will no longer be suitable for radiochemotherapy due to metastatic disease, supporting the view that PET/CT improves target definition and patient selection and should be performed prior to dose planning [31].

Another meta-analysis by Dong et al. determined the efficacy of ^18^F-FDG PET maximum standardised uptake value (SUVmax) in the prediction of tumour responses and the survival of patients with early-stage NSCLC receiving stereotactic body radiotherapy (SBRT). The meta-analysis showed that pre-SBRT primary tumour SUVmax may be prognostic for the outcome of patients with NSCLC treated with SBRT [32].

### 3.4. Gynaecological Cancers

There is an increasing interest in the role of PET/CT in staging and radiation treatment planning in gynaecological cancers and summaries studies shown on table:1 below.

#### 3.4.1. Cervical Cancer

For cervical cancer, the gold standard imaging procedure for treatment planning is the MRI. Although findings from FDG-PET/CT are not part of clinical FIGO staging, various practice guidelines, including the NCCN and the EANM/SNMMI Practice Guideline for Cervical Cancer, recommended adding ^18^FDG PET/CT to the traditional FIGO/TNM staging in baseline assessment and treatment planning, particularly in locally advanced cervical cancer [33,34]. The NCCN 2022 guideline recommended PET/CT for evaluation of stage IB2 or greater disease, especially for evaluation of nodal or extra pelvic tumour, and for post-treatment surveillance (3–6 months post treatment) [33]. The EANM/SNMMI guideline based its recommendation on results from studies demonstrating that ^18^FDG PET/CT is more accurate than pure anatomical imaging in detecting both pelvic and para-aortic nodal metastases in cervical cancer, as well as detecting occult supraclavilar lymph node involvement, and like in other tumour types, ^18^FDG PET/CT-based target volume delineation reduces the inter-observer variability in radiotherapy treatment planning [34]. The use of ^18^FDG-PET/CT for targeted volume delineation and identifying nodal metastasis has been reported in many studies [7,35,36,37]. Tsai et al., in 2004., reported that PET/CT detected additional para-aortic metastases in nearly 30% of patients following baseline CT and MRI [38]. In a systematic review of the literature searching for the evidence for the use of PET for radiation treatment planning in patients with cervical cancer, Salem et al., in 2011, showed that the most accurate imaging technique for identifying nodal metastases in cervical cancer patients was PET-CT [39]. This was shown to impact external-beam radiotherapy planning by modifying the treatment field and customising the radiation dose, particularly in patients with previously undetected para-aortic and inguinal nodal metastases [39]. In a more recent review by Zhou et al. in 2018, 57% of the treatment was modified following a PET/CT examination [40]. Generally, the prominent role of PET/CT treatment planning in cancer of the cervix is in cases of para-aortic lymph node involvement, enlarged pelvic nodes, uterine canal involvement, high tumour grade, and locally advanced disease [35]. See Figure 2 below, in which is shown the axial view of planning CT fused with PET with signal, and which shows FDG uptake on left pelvis node and contoured below. 

#### 3.4.2. Endometrial Carcinoma

PET/MRI may be a useful imaging diagnostic technique for the preoperative staging of endometrial cancer in patients with a high risk of recurrence because it might provide important information for treatment planning, especially if no nodes were sampled [41]. Additionally, it can be used in the planning of brachytherapy for patients who are medically inoperable and in patients with disease recurrences who are candidates for salvage therapy [41]. Particularly in obese patients, PET/MRI has improved accuracy for staging endometrial cancer and may replace lymphadenectomy [36,42].

#### 3.4.3. Ovarian Cancer

The utility of PET/CT in ovarian cancer is still being investigated. A pilot study by Metser et al. showed that ^18^F-DCFPyL-PET had higher specificity than CT in detecting advanced high-grade ovarian serous carcinoma tumour sites. It detects fewer disease sites than CT, especially in the upper abdomen and the gastrointestinal tract, likely limiting its clinical utility [43]. In ovarian cancer, published studies reported the utility of PET/CT in improving the detection of metastatic lymph nodes and recurrent disease. In 2012, Yuan et al. published a meta-analysis carried out to compare the diagnostic performances of computed tomography (CT), magnetic resonance (MR) imaging, and positron emission tomography (PET or PET/CT) for detection of metastatic lymph nodes in patients with ovarian cancer [44]. A total of 18 studies, with 882 patients, were included in the study, which showed that FDG-PET was more accurate than CT or MR imaging in the detection of lymph node metastases [44]. In a 2013 meta-analysis of 29 studies involving 1651 patients Limei et al. reported a pooled sensitivity and specificity of 89% and 90%, respectively, and concluded that PET/CT is a useful tool for predicting the diagnosis and restaging of suspected recurrent ovarian carcinoma [45]. While PET/CT is not routinely used in clinical practice in the treatment of ovarian cancer, the 2022 NCCN guidelines endorses the use of PET/CT for post treatment surveillance and treatment planning for recurrent disease [46].

### 3.5. Anal Cancer

^18^FDG-PET/CT may provide valuable diagnostic information for treatment planning in anal cancer. While MRI and transanal endoscopic ultrasound remains the clinical standard for T-staging, the strength of FDG-PET/CT is the additional detection of small lymph node metastases in unsuspected pelvic and inguinal lymph nodes, and the detection of distant occult metastases [5]. A meta-analysis reported in 2017 showed that PET or PET/CT upstaged 5.1 to 37.5% of patients and down staged 8.2 to 26.7% of patients, and treatment plans were modified in 12.5 to 59.3% of patients [47]. Another study in 2019 reported that up to 20–26% of FDG-PET-positive lymph nodes were located outside the target volume of common practice guideline recommendations for elective lymph node irradiation and would have been missed without the FDG-PET/CT-derived information [48]. Another systematic review and meta-analysis study was performed by Albertson et al. to assess impact on survival, quality of life, symptom score, change in target definition and treatment intention. The meta-analysis found that PET/CT for dose planning results in a change of the target volume for almost one out of every four patients with anal cancer, suggesting that PET/CT imaging will likely benefit patients with anal cancer, justifying its implementation in the routine clinical practice [49]. Indeed the 2022 NCCN guidelines endorsed PET/CT scanning, or PET/MRI if available, to verify staging before treatment, as it has been reported to be useful in the evaluation of pelvic nodes, even in patients with anal canal cancer who have normal-sized lymph nodes on CT imaging [50].

**Table 1 diagnostics-13-00053-t001:** Summary of the studies included for the review.

Article Title	Authors	Type of Article	Comments
The Role of Positron Emission Tomography Imaging in Radiotherapy Target Delineation	Menon, et al. 2020 [6]	Review article	CNS, thoracic, gynaecologic, genitourinary, gynaecologic, hematologic
Use of PET and Other Functional Imaging to Guide Target Delineation in Radiation Oncology	Verma, et al. 2018 [7]	Review article	CNS, thoracic, gynaecologic, genitourinary, gynaecologic, hematologic
Positron emission tomography with computed tomography imaging (PET/CT) for the radiotherapy planning definition of the biological target volume: PART 1	Alongi, et al. 2019 [11]	Expert review	Brain, head and neck and lungs
Response Assessment in Neuro-Oncology working group and European Association for Neuro-Oncology recommendations for the clinical use of PET imaging in gliomas	Albert, et al. 2016 [15]	Practice Guideline Practice	Brain
Contribution of PET imaging to radiotherapy planning and monitoring in glioma patients—a report of the PET/RANO group	Galldiks, et al. 2021 [17]	Practice Guideline	Brain
Role of 18FDG-PET/CT for radiotherapy planning in head and neck cancer	Farre, et al. 2013 [23]	Cross-sectional	Head and neck
Tumour volume in pharyngolaryngeal squamous cell carcinoma: comparison at CT, MR imaging, and FDG PET and validation with surgical specimen	Daisne, et al. 2004 [24]	Cross-sectional	Head and neck
Variation in Radiotherapy Target Volume Definition, Dose to Organs at Risk and Clinical Target Volumes using Anatomic (Computed Tomography) versus Combined Anatomic and Molecular Imaging (Positron Emission Tomography/Computed Tomography): Intensity-modulated Radiotherapy Delivered using a Tomotherapy Hi Art Machine: Final Results of the Vortig ERN Study	Chatterjee, et al. 2012 [25]	Cross-sectional	Head and neck
Impact of the type of imaging modality on target volumes delineation and dose distribution in pharyngo–laryngeal squamous cell carcinoma: comparison between pre- and per-treatment studies	Geets, X, et al. 2006 [26]	Cross-sectional	Head and neck
Combined 18F-FDG-PET/CT Imaging in Radiotherapy Target Delineation for Head-and-Neck Cancer	Guido, et al. 2009 [27]	Cross-sectional	Head and neck
What is the prognostic impact of FDG PET in locally advanced head and neck squamous cell carcinoma treated with concomitant chemo-radiotherapy? A systematic review and meta-analysis.	Bonomo P, et al. 2018 [28]	Systematic review	Head and neck
18F-FDG-PET in guided dose-painting with intensity modulated radiotherapy in oropharyngeal tumours: A phase I study (FiGaRO).	Michaelidou A, et al. 2021 [29]	Phase 1 trail	Head and neck
Perspective paper about the joint EANM/SNMMI/ESTRO practice recommendations for the use of 2-[18F] FDG-PET/CT external beam radiation treatment planning in lung cancer	Vaz, et al. 2022 [30]	Guideline perspective	Lung
Positron emission tomography and computed tomographic imaging (PET/CT) for dose planning purposes of thoracic radiation with curative intent in lung cancer patients: A systematic review and meta-analysis.	Hallqvist A, et al. 2017 [31]	Systematic review	Lung
Prognositc significance of SUVmax on pretreatment 18 F-FDG PET/CT in early-stage non-small cell lung cancer treated with stereotactic body radiotherapy: A meta-analysis	Dong M, et al. 2017 [32]	Meta-analysis	Lung
NCCN clinical practice guidelines in oncology: cervical cancer version 1	NCCN 2022 [33]	Practice guideline	Cervix
[18F] FDG-PET or PET/CT in the evaluation of pelvic and para-aortic lymph nodes in patients with locally advanced cervical cancer: A systematic review of the literature	Adam, et al. 2020 [34]	Systematic review	Cervix
Positron emission tomography with computed tomography imaging (PET/CT) for the radiotherapy planning definition of the biological target volume: PART 2	Fiorentino, et al. 2019 [35]	Expert review	Pancreas, prostate, gynaecological and rectum/anal cancer.
The role of PET-CT in radiotherapy planning of solid tumours	Jelercic, et al. 2015 [37]	Review article	Lung, head and neck, oesophageal and cervix
Evidence for the use PET for radiation therapy planning in patients with cervical cancer: a systematic review	Salem, et al. 2011 [39]	Systematic review	Cervix
The Role of PET Imaging in Gynaecologic Radiation Oncology.	Rao YJ, et al. 2018 [41]	Review	Gynaecological
Computer tomography, magnetic resonance imaging, and positron emission tomography or positron emission tomography/computer tomography for detection of metastatic lymph nodes in patients with ovarian cancer: A meta-analysis.	Yuan Y, et al. 2012 [44]	Meta-analysis	Ovary
NCCN clinical practice guidelines in oncology: Ovarian Cancer/Fallopian Tube Cancer/Primary Peritoneal Cancer, version 5	NCCN 2022 [46]	Practice guideline	Ovary
Prostate-specific membrane antigen PET-CT in patients with high-risk prostate cancer before curative-intent surgery or radiotherapy (proPSMA): a prospective, randomized, multicentre study	Hofman, et al. 2020 [47]	RCT, cross over trial	Anal
Positron emission tomography and computed tomographic (PET/CT) imaging for radiation therapy planning in anal cancer: A systematic review and meta-analysis	Albertsson P, et al. 2018 [49]	Systematic review	Anal
PET/CT in Radiation Therapy Planning	Specht, et al. 2018 [51]	Review article	Lymphoma
PET imaging in anal canal cancer: a systematic review and meta-analysis	Mahmud, et al. 2017 [52]	Systematic review	Anal

### 3.6. Prostate Cancer

In treatment planning for prostate cancer, detecting intra- and extra-prostatic tumour foci is essential. A review of the value of PET imaging in radiation therapy reported that most studies in the last five years showed that PET imaging with a prostate-specific membrane antigen (PSMA) tracer allows more sensitivity and specificity in the detection of prostate cancer lesions than other imaging techniques [5]. Hoffman et al., in 2020, reported the results of the proPSMA study, a phase III randomised trial on whether novel imaging using prostate-specific membrane antigen (PSMA) PET-CT might improve accuracy and affect management. They reported that the application of PSMA-PET/CT has higher accuracy for detecting lymph nodes and bone metastases than CT or bone scan, therefore, having a relevant impact on patient management [51]. Karagiannis et al., in a study of 43 patients, showed that PSMA-PET/CT influences RT choice, adjuvant therapy regimes, and treatment decisions in the treatment of prostate cancer [52]. In that study, the final treatment choice was affected by PSMA-PET/CT outcome in 60.5% of cases, and only 39.5% of the patients who underwent PSMA-PET/CT were treated according to their initial treatment plans [52]. More clinical trials are underway to concretise the role of PET/CT in prostate cancer.

### 3.7. Others

In lymphomas, PET/CT provides essential information for planning of RT, allowing optimal target coverage while minimizing unnecessary irradiation of normal tissues. Most lymphomas have avid uptake of FDG, making it the main tracer used in PET in lymphomas [53].

In rectal cancer, the role of PET/CT in planning therapy has not been established, although it may be helpful in target delineation. However, it is a promising tool for diagnostic workups [35].

In pancreatic cancer, FDG-PET has helped in the differential diagnosis between pancreatitis and adenocarcinomas and can also be used to detect distant metastasis. It can become a tool for the definition of the target volume [35].

## 4. PET/CT Services in Africa

South Africa and North Africa are confirmed to have the highest density of clearly established, nuclear medicine departments, but the PET-CT treatment and diagnostic approach has been underutilised in sub-Saharan Africa [52]. This approach has been available in South Africa for more than 15 years and is now a mature imaging modality in clinical medicine. South Africa currently has 21 PET-CT cameras, with five (24%) in the public sector (managing at least 82% of the country’s patients), 13 (62%) in the private sector (serving 18% of the country’s patients), and 3 (14%) in dedicated research units. These institutions have access to two commercially available radiotracers with oncological applications, excluding research tracers that are compounded on-site: FDG and FDOPA [54].

On the other hand, private companies are hesitant to invest in the sub-Saharan countries due to the weak economy and small market [55], as well as due to shortages of qualified personnel and radiotracers [55,56], which are some of the challenges confronting the PET-CT approach’s implementation in sub-Saharan African countries (Table 1).

## 5. Conclusions

In conclusion, PET/CT imaging for the guidance of radiotherapy planning has come to stay and will likely expand to include more cancer types. PET/CT particularly paves the way for personalised treatment planning in oncology and enables increased efficacy with decreased side effects to the organs at risk. While it is an established part of planning treatment for some cancers, multicenter and prospective trials are needed to enhance evidence for its routine implementation, together with continued development of improved tracers. It is important to note that sub-Saharan Africa (excluding South Africa) is lagging in this important leap in cancer management.

## Figures and Tables

**Figure 1 diagnostics-13-00053-f001:**
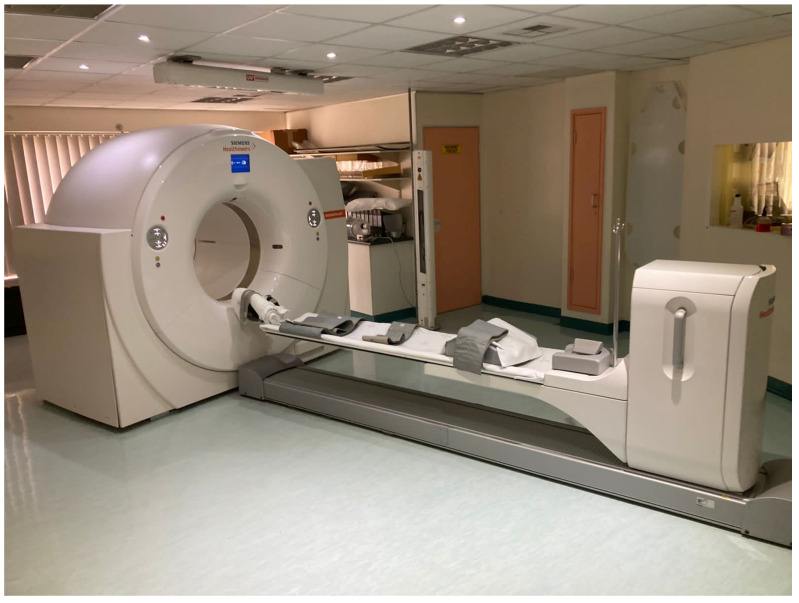
PET/CT with capacity of radiotherapy treatment planning; anatomical and metabolic images are acquired with the patient in the same position.

**Figure 2 diagnostics-13-00053-f002:**
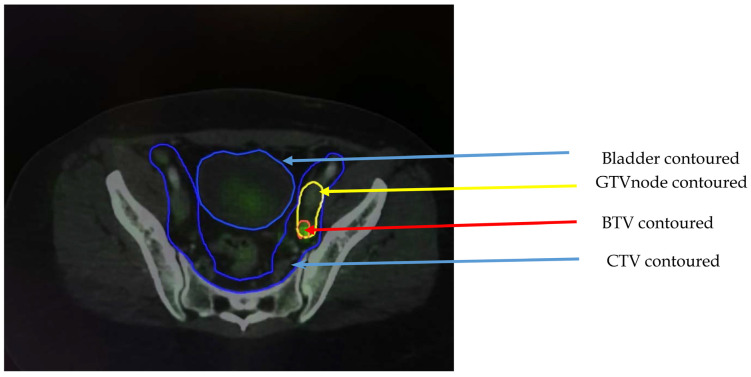
FIGO stage IIIC cervix cancer: Axial view of the PET/CT treatment planning with the label: Bladder (organ at risk), GTVn (Gross tumour volume-node contoured on CT), BTV (Biological tumour volume contoured on PET treatment planning) and CTV (Clinical tumour volume was contoured on CT planning).

## Data Availability

Not applicable.

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
