# Peer review of "The Role of Positron Emission Tomography and Computed Tomographic (PET/CT) Imaging for Radiation Therapy Planning: A Literature Review"

_diagnostics, 2022, doi:10.3390/diagnostics13010053_

Round 1

Reviewer 1 Report

Thank you very much for the opportunity to review this work. The manuscript is well written, but I have some comments to provide to the authors, in order to improve the quality of this paper.

1) in the paragraph "Principles of18F-FDG-PET", please explain why 18F is the most used isotope for clinical purposes. You can also add a table with the most important physical properties of the isotopes that you cited in the manuscript.

2) in the paragraph "Central Nervous System (Brain) Cancers", please explain better the clinical role of  amino acid radiopharmaceuticals in clinical practice and indications. 

3)in the paragraph "Prostate Cancer" you can also add some sentences regarding Choline PET. 

Author Response

Dear Reviewer 

Thank you for the opportunity the comments and please kindly find the attached response to all the comments. 

Regards 

Abbas

Reviewer 2 Report

Generally, this is a well-written manuscript. The author gave a detailed introduction and background. The role of PET in radiotherapy is easy to follow and understand. The author listed several different types of cancer that may be benefit from the PET/CT. All of that make sense and reasonable. However, the authors didn't generate any statistical analysis. The conclusion part, the author mentioned "PET/CT may increase the efficacy with decrease sided effect to the organs at risk". Without the statistical analysis support, this will be hard for readers to fully understand this. I strongly suggest the authors put some statistical analysis such as univariate/descriptive analysis in the future study. 

Author Response

Dear Reviewer 

Thank you for the comments and we highly appreciate and on the futures article we will back literature with statistics before reaching to conclusions. 

Millions thanks 

Regards 

Abba Mallum 

MB'BS, FC RAD ONC(SA), MMED(Stell)